# Probiotic Ingestion, Obesity, and Metabolic-Related Disorders: Results from NHANES, 1999–2014

**DOI:** 10.3390/nu11071482

**Published:** 2019-06-28

**Authors:** Eva Lau, João Sérgio Neves, Manuel Ferreira-Magalhães, Davide Carvalho, Paula Freitas

**Affiliations:** 1Department of Endocrinology, Diabetes and Metabolism, Centro Hospitalar Univesrsitário São João, 4200 Porto, Portugal; 2CINTESIS—Centre for Health Technologies and Information Systems Research, Faculty of Medicine, University of Porto, 4200 Porto, Portugal; 3Department of Surgery and Physiology, Cardiovascular Research Centre, Faculty of Medicine, University of Porto, 4200 Porto, Portugal; 4Health Information and Decision Sciences Department, Faculty of Medicine, Porto University, 4200 Porto, Portugal; 5I3S—Instituto de Investigação e Inovação em Saúde, Faculty of Medicine, University of Porto, 4200 Porto, Portugal

**Keywords:** intestinal microbiota, probiotics, nutrients

## Abstract

Gut microbiota dysbiosis has been recognized as having key importance in obesity- and metabolic-related diseases. Although there is increasing evidence of the potential benefits induced by probiotics in metabolic disturbances, there is a lack of large cross-sectional studies to assess population-based prevalence of probiotic intake and metabolic diseases. Our aim was to evaluate the association of probiotic ingestion with obesity, type 2 diabetes, hypertension, and dyslipidemia. A cross-sectional study was designed using data from the National Health and Nutrition Examination Survey (NHANES), 1999–2014. Probiotic ingestion was considered when a subject reported consumption of yogurt or a probiotic supplement during the 24-h dietary recall or during the Dietary Supplement Use 30-Day questionnaire. We included 38,802 adults and 13.1% reported probiotic ingestion. The prevalence of obesity and hypertension was lower in the probiotic group (obesity-adjusted Odds Ratio (OR): 0.84, 95% CI 0.76–0.92, *p* < 0.001; hypertension-adjusted OR: 0.79, 95% CI 0.71–0.88, *p* < 0.001). Accordingly, even after analytic adjustments, body mass index (BMI) was significantly lower in the probiotic group, as were systolic and diastolic blood pressure and triglycerides; high-density lipoprotein (HDL) was significantly higher in the probiotic group for the adjusted model. In this large-scale study, ingestion of probiotic supplements or yogurt was associated with a lower prevalence of obesity and hypertension.

## 1. Introduction

Obesity is a pro-inflammatory state that plays a central role in the progression of several diseases, such as type 2 diabetes, hypertension, and dyslipidemia [1]. The pathophysiology of obesity and metabolic-related diseases is complex, resulting from the imbalance between environmental and genetic factors. The human gastrointestinal tract is populated by a complex ecosystem—the gut microbiota—which is responsible for the regulation of essential functions for the maintenance of health, including protective, structural, and histological functions [2]. New insights emphasize the role of gut microbiota in energy homeostasis, giving rise to the “The Metagenome Hypothesis” as a key player in the comprehension of metabolic diseases [3,4,5]. Accordingly, recent studies have shown the relationship between intestinal dysbiosis, which is defined as a change in the composition of gut microbiota and glucose and lipid metabolism deregulation in obesity and type 2 diabetes [6,7]. Obesity has been linked to an increase in Firmicutes and a decrease in Bacteroidetes [5,8]. Likewise, two large metagenome-wide association studies reported that type 2 diabetes had a lower proportion of butyrate-producing *Clostridiales* (*Roseburia* and *Faecalibacterium prausnitzii*) and greater proportions of *Clostridiales* that do not produce butyrate [7,9]. On one hand, distinct differences in gut microbiota result in a greater increase in harvesting energy from the diet by fermentation and the absorption of dietary polyssacharides, promoting hepatic lipogenesis. On the other hand, gut microbiota regulate intestinal permeability and an increase in the translocation of lipopolysaccharide-containing gut microbiota increases the inflammatory state, which is named metabolic endotoxemia, accompanied by weight gain and insulin resistance [10]. Understanding this interplay between gut microbiota and the host has created interest in shaping microbiota to prevent, treat, or delay obesity, type 2 diabetes, and metabolic-associated complications.

Probiotics are food components or supplements with living microorganisms that confer health advantages to the host [11]; specific strains have been increasingly studied as a potential therapeutic approach to shape gut microbiota composition, with possible benefits for weight control and diabetes management [12,13,14,15]. The administration of probiotics may restore the crosstalk between human host and gut microbiota, controlling homeostatic functions during obesity and metabolic-related disorders. Although there is increasing evidence of the potential benefits of probiotics in metabolic diseases, there is a lack of large cross-sectional studies to evaluate the population-based prevalence of probiotic intake and metabolic differences in those exposed to probiotics compared to those who are not. Large population surveys can create powerful information about population health status and trends. To date, there are no published studies about the large-scale use of probiotic supplements and yogurts and possible associations with metabolic diseases. This type of analysis can produce high-quality data for the real-life use of these types of food and supplements. Our aim was to assess the association of probiotic ingestion, through yogurt or supplements, with the prevalence of obesity and associated metabolic disturbances, namely type 2 diabetes, hypertension, and dyslipidemia.

## 2. Materials and Methods

### 2.1. Study Design and Settings

We designed a cross-sectional analysis, using data from the National Health and Nutrition Examination Survey (NHANES). NHANES is a national research survey designed to collect demographic, socio-economic, health, and nutritional statuses from a representative sample of the non-institutionalized civilian resident population of the United States of America. NHANES is a major program of the National Centre for Health Statistics (NCHS), which is part of the Centers for Disease Control and Prevention (CDC), and the detailed methodology is described in the literature [16]. NHANES was approved by the NHANES Institutional Review Board (IRB) and the NCHS Research Ethics Review Board (ERB) (after 2003).

### 2.2. Participants and Data Collection

We included adults aged 18 years or older, who had been included in NHANES between 1999 and 2014. Pregnant women were excluded. NHANES participants without physical examination or laboratory data and with no dietary data or implausible dietary data (24-h dietary recall) were also excluded. Appendix A shows the flowchart of the study population. NHANES data collection was performed through an in-home interview for demographic and basic health information data collection, together with a health examination in a Mobile Examination Centre (MEC), where participants were examined and surveyed. NHANES MEC examinations included anthropometric measurements, blood pressure assessment, and blood workup. Data were collected by a trained interviewer who had completed an intensive training course administered by the US Department of Agriculture and the US Department of Health and Human Services.

### 2.3. Assessment and Definition of Probiotic Exposure

In all NHANES cycles, from 1999 to 2014, a 24-h dietary recall was collected. Using an automated multiple-pass method, a detailed dietary intake (quality and quantity) for the 24-h period before the interview was recorded. For participants in the 1999–2002 NHANES, only one in-person 24-h dietary recall was performed. From 2003 onward, an additional telephone dietary recall interview was also performed 3 to 10 days following the in-person dietary interview. For the participants of the 2003–2014 NHANES, we used the mean of the nutritional information from both recalls (in-person recall and telephone recall). To assess probiotic supplementation exposure, we also used the Dietary Supplement Use 30-Day (DSQ), which assesses food supplement use during the preceding 30 days. Appendix A lists the probiotic supplements included.

Probiotic ingestion was considered when a subject reported consumption of a probiotic supplement or yogurt (as a dietary source of probiotics) during the 24-h dietary recall or of a probiotic supplement during the DSQ. Non-yogurt foods containing probiotics were classified as probiotic supplements. 

### 2.4. Definition of Metabolic Comorbidities, Smoking, and Physical Activity

Obesity was defined as a body mass index (BMI) ≥30 Kg/m^2^. Type 2 diabetes was defined as glycated hemoglobin (HbA1c) ≥6.5%, fasting plasma glucose level ≥126 mg/dL, or current glucose-lowering drug use. Dyslipidemia was assumed if participants had low-density cholesterol (LDL) ≥160 mg/dL, high-density cholesterol (HDL) <40 mg/dL, triglycerides ≥200mg/dL, total cholesterol ≥240 mg/dL, or if they were being treated with lipid-lowering drugs. Systolic and diastolic blood pressures (BP) were determined by the mean of 3 or 4 consecutive blood pressure readings. Hypertension was defined as systolic blood pressure ≥140 mmHg, or diastolic blood pressure ≥90 mmHg, or current medication for hypertension. 

Smoking status was classified as former and current smokers. Current smokers were those who reported smoking at least 100 cigarettes during their lifetime and were currently smoking every day, or some days. Former smokers were those who reported smoking at least 100 cigarettes during their lifetime, but do not currently smoke.

Physical activity was measured differently along NHANES cycles. We classified participants using variables that allowed categorization of physical activity level into three categories (low, intermediate, and high). From 1999 to 2006 the physical activity level was assessed with the question “compare activity with others of the same age” (participants answering “less active” were classified into category “low”, “about the same” into category “intermediate”, and “more active” into category “high”). From 2007 to 2014, the weekly metabolic equivalents (MET) minutes of physical activity (accounting for vigorous work-related activity, moderate work-related activity, walking or bicycling for transportation, vigorous leisure-time physical activity, and moderate leisure-time physical activity) was divided into tertiles (participants were classified as “low” if included in the lower MET-minute tertile, as “intermediate” if in the middle MET-minute tertile, and as “high” if in the higher MET-minute tertile).

### 2.5. Statistical Analysis

Statistical analysis took into account the complex survey design of the NHANES dataset and was performed according to the CDC analytic recommendations [17].

Continuous variables were described as mean ± standard deviation (SD) and categorical variables were described as absolute and relative frequencies. To assess the association between probiotic exposure and metabolic comorbidities (obesity, diabetes, hypertension, and dyslipidemia), we performed unadjusted and adjusted logistic regression models. To evaluate the association between probiotic exposure and cardiomatebolic parameters (BMI, HbA1c, fasting plasma glucose, systolic BP, diastolic BP, LDL, HDL, and triglycerides), we performed unadjusted and adjusted multivariate linear regression models. We excluded those participants who were being treated with anti-hypertensive drugs from systolic and diastolic BP analysis, participants treated with anti-dyslipidemic drugs from analysis concerning lipid profiles, and patients treated with antidiabetic drugs from HbA1c and fasting plasma glucose analysis.

In the adjusted analyses, we used the following models (Appendix A): Model 1, including age, sex, ethnicity (Mexican American, other Hispanic, non-Hispanic white), annual family income (<$25,000, $25,000 to $75,000, >$75,000), and education (<9th grade, ≥9th grade); model 2, including all model 1 covariates plus alcohol intake, smoking status (never a smoker, current smoker, or former smoker), physical activity (low, intermediate, high), ingested kcal per day, ingested carbohydrates/kcal per day, ingested protein/kcal per day, ingested fiber/kcal per day, and ingested polyunsaturated/saturated fatty acids ratio. In model 2, we also included BMI in all analyses except in the obesity analysis and sodium intake per day only in the hypertension and blood pressure analyses. As a supplementary analysis, we performed an additional model (model 3) that classified the diet pattern using the Dietary Approaches to Stop Hypertension (DASH) score. In model 3 we included all model 1 covariates plus alcohol intake, smoking status, physical activity, and the DASH dietary pattern score. Model 3 also included BMI in all analyses except the obesity analysis and sodium intake per day only in the hypertension and blood pressure analyses. The DASH score is based on 9 target nutrients (sodium, total fat, saturated fat, protein, fiber, cholesterol, calcium, magnesium, and potassium), as previously described [18]. Individuals meeting the DASH goal were given a score of 1.0 for that nutrient; if they attained an intermediate goal, they were given a score of 0.5 for that nutrient. The DASH score is the sum of the score for each individual nutrient. Logistic regression results were expressed as an odds ratio (OR) and a 95% confidence interval (95% CI). A two-sided *p*-value of <0.05 was considered statistically significant. Analyses were performed with Stata (version 14.2).

## 3. Results

### 3.1. Baseline Characteristics According to Probiotic Consumption

We included 38,802 adults, of whom 13.1% had exposure to probiotic supplements or yogurt. Baseline population characteristics according to probiotic consumption are described in Table 1. Participants in the group exposed to probiotic supplements or yogurt were more likely to be female, older, non-Hispanic white and to have a higher income and education level. Ingestion of kcal/day was similar between groups.

### 3.2. Prevalence of Metabolic Comorbidities According to Probiotic Consumption

The prevalence of metabolic comorbidities according to probiotic supplement or yogurt exposure is represented in Figure 1. All four studied comorbidities were lower in the exposed group. The comorbidities that showed higher differences in prevalence between groups were obesity and hypertension; 5.4% and 2.9% lower, respectively. Diabetes prevalence difference was only 1.6%, but this was still significant in our analysis.

### 3.3. Modulation of Metabolic Comorbidities According to Probiotic Consumption

Table 2 summarizes the results of metabolic disturbances by using unadjusted and adjusted models of the prevalence of comorbidities according to probiotic supplement or yogurt exposure. For unadjusted analysis, participants exposed to probiotics manifested a 22% reduction in the odds of having obesity (OR: 0.78, 95% CI 0.71–0.86; *p* < 001), a 16% reduction in the odds of having diabetes (OR: 0.84, 95% CI 0.73–0.96; *p* = 0.020), and a 12% reduction in the odds of having hypertension (OR: 0.88, 95% CI 0.81–0.96; *p* = 0.004). No significant differences were found in dyslipidemia prevalence.

After adjusting for potential confounders, obesity and hypertension prevalence remained significantly lower in the probiotic exposed group (Table 2). In turn, diabetes prevalence became similar between groups. 

Table 3 summarizes the results of cardiometabolic parameters according to probiotic supplement or yogurt exposure. In the unadjusted analysis, all the studied markers were lower in the probiotic-exposed group, with the exception of HDL (which was higher in the exposed group) and LDL (no differences were seen). After adjusting for potential confounders, BMI, and systolic BP, diastolic BP and triglycerides remained significantly lower in the probiotic-exposed group and HDL remained significantly higher (Table 3). Appendix A summarize the odds ratios of disease and the variation of cardiometabolic parameters, respectively, according to probiotic supplement or yogurt exposure, after accounting for the DASH dietary pattern score. The associations of probiotic supplement or yogurt ingestion with cardiometabolic parameters were not different after adjusting for DASH diet adherence.

The odds of metabolic comorbidities according to the origin of probiotics is presented in Figure 2. Among those participants exposed to probiotics, 95.7% were exposed to yogurt and 5.4% were exposed to probiotic supplements. The odds of obesity, diabetes, hypertension, and dyslipidemia for participants exposed to yogurt or to probiotic supplements alone were similar to the odds for participants exposed to any type of probiotic. Although the confidence intervals were wider in the analysis of probiotic supplements (due to the smaller number of exposed individuals), the point estimates for the association with comorbidities was similar to the association with exposure to any probiotic or exposure to yogurt, with the exception of diabetes. Although the association in both analyses was not significant, the odds ratio for diabetes was 0.92 (95% CI 0.77–1.10) among participants exposed to yogurt and 1.28 (95% CI 0.71–2.23) among those exposed to probiotic supplements. 

## 4. Discussion

We conducted a cross-sectional analysis on a large and representative US population, for a total of 38,802 adults, and found that 13.1% reported the use of probiotic supplements or yogurt ingestion. Although there are several studies addressing the possible beneficial associations of probiotic ingestion and several metabolic outcomes, there is a lack of large cross-sectional studies to objectively assess population-based prevalence of probiotic intake and metabolic differences in those exposed and not exposed to probiotics. To our knowledge, this was the first large cross-sectional analysis aiming to assess the association of probiotic ingestion, either by probiotic supplements or yogurt, with metabolic disturbances. Probiotic ingestion was associated with a 17% lower prevalence of obesity and a 21% lower prevalence of hypertension. Furthermore, HDL cholesterol was significantly higher and triglyceride levels were significantly lower in the probiotic group.

Probiotics modulate gut microbial communities, exerting beneficial metabolic effects through the regulation of multitudinous physiological metabolic pathways. Among the molecular mechanisms, the regulation of adipogenesis, stimulation of insulin signaling, improvement of gut barrier function, reduction of metabolic endotoxemia, and down-regulation of cholesterol levels are some of the suggested key players in the crosstalk between probiotics and metabolic disorders [19].

### 4.1. Obesity

We found that probiotic ingestion, via supplements or yogurt, was associated with a lower prevalence of obesity (17% reduction), before and after adjusting for demographics and potential confounders. Although subjects that consumed probiotics had a higher consumption of carbohydrates, fiber, and protein, the effects of probiotics on BMI were significant, even after adjustment for confounders (−0.41 Kg/m^2^ between groups in model 2). Furthermore, there were no differences in total energy intake per day between groups and, also, there was no linear association of physical activity level (low, intermediate, and high) and probiotic ingestion. The only thing observed was a higher proportion of intermediate physical activity level, but lower proportions of high and low physical activity levels in the probiotic-exposed group. Putting this all together, these results support our hypothesis of the beneficial impact of probiotic ingestion per se on metabolic health, namely its effects on the regulation of body weight. In agreement with our study, a recent meta-analysis of randomized clinical trials with 957 subjects, with a mean BMI of 27.6 kg/m^2^, showed that probiotic administration significantly reduced body weight by −0.60 kg and BMI by −0.27 kg/m^2^ [20]. NHANES’s data between 1999–2004 also showed that yogurt consumption was associated with a lower likelihood of having obesity (OR: 0.57, 95% CI 0.40–0.82; *p* < 0.05) [21]. A prospective study including 120,877 US individuals evaluated lifestyle factors and weight change at four-year intervals, with multivariable adjustments. The four-year weight change was negatively associated with yogurt ingestion [22], which further supports our results.

### 4.2. Diabetes

In the unadjusted analysis, probiotic supplement and yogurt consumers had lower odds of having diabetes and, accordingly, lower glycemia and HbA1c levels. However, when adjusted for individual characteristics and confounders, the difference was no longer significant. One possible explanation for this is that diabetes is largely determined by individual demographic characteristics, such as age, ethnicity and, mainly, BMI [23]. Therefore, there are no longer differences after adjusting for individual factors. Furthermore, in another meta-analysis of randomized controlled trials, Ruan I et al. [14]. concluded that probiotics had a greater effect on fasting blood glucose for people with diabetes. On the contrary, those without diabetes only show a trend of a glucose-lowering effect, which shows that probiotic supplementation may have a greater benefit for individuals with higher fasting glucose levels [14]. In three prospective cohorts in the US, yogurt intake was consistently and inversely associated with type 2 diabetes risk [24]. Probiotic supplementation seems to be more effective in reducing HbA1c in diabetic patients with higher baseline BMIs and, furthermore, probiotic supplements with greater bacterial species may be more effective [25], which we could not evaluate in our study. Yao K et al. [26] conducted a meta-analysis in patients with type 2 diabetes to investigate the effects of probiotics on glucose metabolism, and, similar to our results, they did not find a significant effect on fasting blood glucose levels, although HbA1c was improved with probiotic supplementation. In the analysis according to the origin of probiotics, we found a non-significant trend to higher odds of diabetes in participants exposed to probiotic supplements, which was not seen in participants exposed to yogurt. This may be explained by a reverse-causality relationship. Participants with diabetes may be more prone to consuming probiotic supplements due to their known potential to improve glucose control in diabetes.

### 4.3. Dyslipidemia

We did not find differences in the odds of dyslipidemia according to probiotic ingestion, however, we did observe some beneficial aspects in the lipid profile of the probiotic-exposed group. HDL was significantly increased and triglycerides were significantly decreased in the probiotic group, even after adjustment. Our results are in line with a study by Fu et al. [27], with 893 subjects, which showed that gut microbiota was associated with a 4.5% variance in BMI, a 6% variance in blood levels of triglycerides, and a 4% variance in HDL, but had little effect on LDL or total cholesterol. In contrast, a meta-analysis including 1624 participants (828 in the probiotic and 796 in the placebo group) demonstrated that probiotics reduced total cholesterol and LDL cholesterol by 7.8 mg/dL and 7.3 mg/dL, respectively but had no significant effects on HDL cholesterol or triglycerides [28]. Human clinical studies have yielded different results on the association of lipid profiles with probiotic supplementation. Differences in the type of probiotics and in the experimental designs, including the clinical heterogeneity of participants, namely their baseline levels of blood lipids, may affect the role of probiotics in lipid metabolism and explain the different results. Given the clinical correlation between obesity and related metabolic disorders, it is possible that the observed associations between gut bacterial composition and lipid levels can be mediated, in part, through the effects on BMI.

### 4.4. Hypertension

Ingestion of probiotic supplements or yogurt resulted in a 21% reduction in the odds of hypertension after adjusting for potential confounders. Both systolic BP and diastolic BP were significantly lower. A meta-analysis of randomized clinical trials supports our results, showing that probiotic consumption significantly reduced systolic BP by −3.56 mmHg and diastolic blood pressure by −2.38 mmHg, compared with control groups [29]. In our study, for the adjusted model, probiotic ingestion was associated with a lower systolic BP by 1.48 mmHg and a lower diastolic BP by 0.86 mmHg. The modulation of BP by probiotics may be linked to several mechanisms, including their capacity to improve lipid profiles, to reduce BMI, and to produce bioactive peptides with angiotensin-converting inhibitory activity [30,31,32].

### 4.5. Strengths and Limitations

Our study has a number of strengths and limitations, which need to be highlighted. We conducted an analysis of a large cross-sectional survey, which was representative of the US population. To the best of our knowledge, this is the first cross-sectional study aiming to assess the association of probiotic ingestion, through supplements or yogurt, and metabolic disturbances. A previous study on the NHANES population was carried out, aiming to evaluate the association between dairy products (which included yogurt ingestion) with obesity and other disturbances of metabolic syndrome; however it did not evaluate probiotic supplementation [21]. We also developed an analytic strategy that included adjusted logistic regression models to obviate confounders such as physical activity, alcohol consumption, and smoking status. One of the limitations of this type of analysis is that we cannot deduce causation; however, we were able to show strong associations between probiotic ingestion and the prevalence of some metabolic disturbances. The participants in the probiotic-exposed group were more likely to be non-Hispanic white. It has been previously stated that gut microbiome varies by geographic ancestry [33], which may limit the extrapolation of these results to other ethnic groups. Furthermore, inter-individual differences in the composition of gut microbiota were previously associated with different responses to probiotics [34], including non-responders to gut microbiota modulation. The absence of individual gut microbiota analyses in our study may have hampered the analysis of these type of responders. The assessment of probiotic exposure was based on self-reported information, however, NHANES only provides dietary information that is considered to be reliable [35]. In our study, we defined probiotic ingestion as being either yogurt or probiotic supplement consumption. Assuming that there could be differences between the types of ingestion, we performed a sensitivity analysis based on the origin of probiotics. There were no significant differences in associations according to the source of exposure to probiotics. Probiotic supplements and yogurts vary in the amounts of bacteria per serving they are composed of. In our study, the population was classified according to whether or not they were exposed to probiotic supplements or yogurt. The duration and quantity of exposure were not taken into account, which may have diluted the magnitude of the association between probiotic consumption and metabolic disturbance.

In summary, our study supports the beneficial association of probiotic supplement or yogurt ingestion with metabolic health, specifically obesity and hypertension. Furthermore, probiotic ingestion was significantly associated with higher HDL cholesterol and lower triglyceride levels. Our study supports the possibility of gut microbiota modulation by the use of probiotics as an attractive therapeutic target to prevent and treat obesity and related cardiometabolic disorders. Future research should focus on understanding the gut microbiota ecosystem and on identifying individuals who benefit the most from selective modulation of microbiota.

## Figures and Tables

**Figure 1 nutrients-11-01482-f001:**
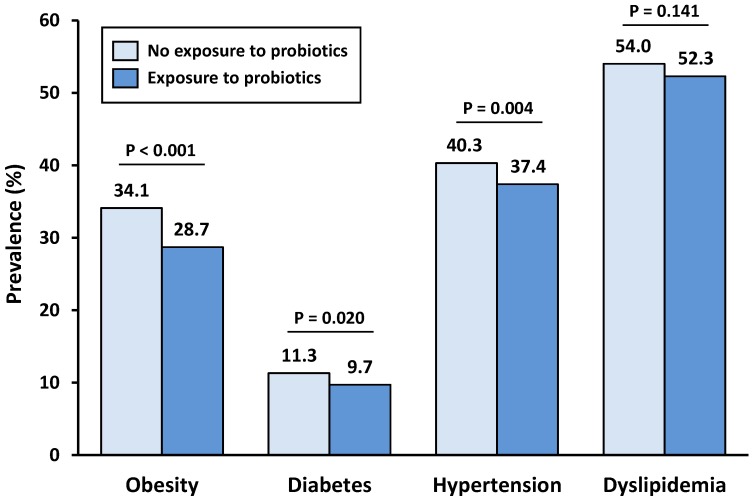
Prevalence of obesity, diabetes, hypertension, and dyslipidemia, according to probiotic exposure.

**Figure 2 nutrients-11-01482-f002:**
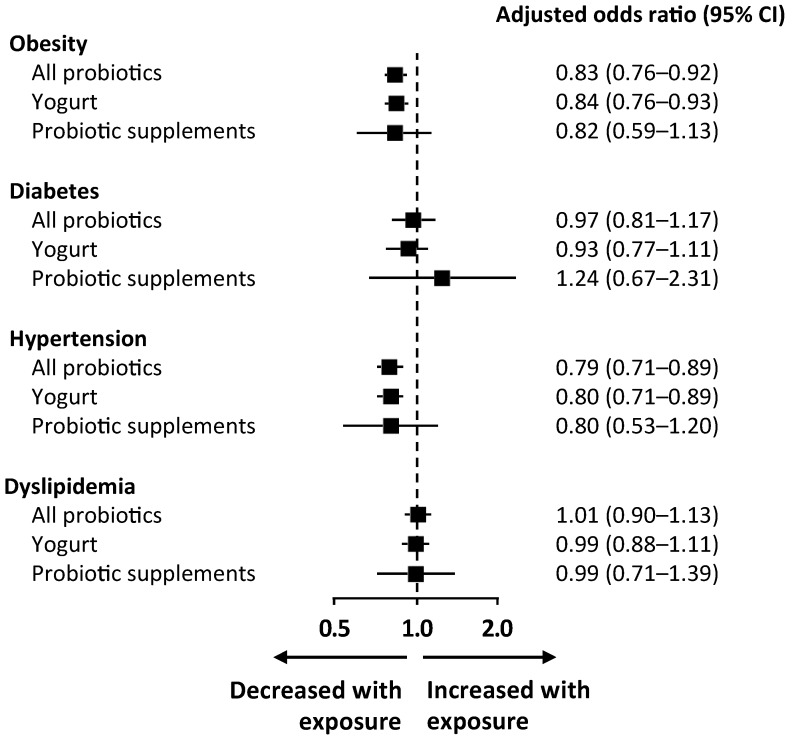
Odds ratio of disease in participants exposed to probiotics compared to those not exposed, according to the origin of probiotics (all probiotics, yogurt, or probiotic supplements). Logarithmic regression models adjusted for age, sex, race, income, education, alcohol intake, smoking status, physical activity, carbohydrates/kcal per day, protein/kcal per day, fiber/kcal per day, and polyunsaturated/saturated fatty acids ratio (model 2). Model 2 also includes BMI in all analyses except in the obesity analysis and includes sodium intake per day only in the hypertension analysis.

**Table 1 nutrients-11-01482-t001:** Baseline population characteristics according to probiotic consumption (*n* = 38,802).

	No Exposure to Probiotics	Exposure to Probiotics	*p* Value
Participants, n (%)	33,719 (86.9%)	5083 (13.1%)	n.a.
Socio-economic characteristics			
Male gender, %	50.2 %	35.0 %	<0.001 *
Age, years ± SD	46.0 ± 15.3	48.9 ± 13.2	<0.001 *
Annual family income <$25000, %	30.7%	19.6%	<0.001 *
Education level less than 9th grade, %	6.4%	2.7%	<0.001 *
Ethnicity			<0.001 *
Non-Hispanic White, %	68.4%	79.5%	
Non-Hispanic Black, %	11.9%	5.4%	
Mexican American, %	8.5%	5.0%	
Other Hispanic, %	5.2%	4.1%	
Other ethnicities, %	6.0%	5.9%	
Risk factors			
Current smokers, %	20.2%	7.6%	<0.001 *
Former smokers, %	28.3%	31.7%	0.002 *
Alcohol consumption >20 g/day, %	15.7%	13.9%	0.028 *
Physical activity level ^#^			0.028 *
Low	28.8%	26.9%	
Intermediate	36.6%	39.3%	
High	34.6%	33.8%	
Nutritional characteristics			
Kcal/day, kcal ± SD	2060.7 ± 648.9	2042.4 ± 529.3	0.163
Carbohydrates/day, g/100 kcal ± SD	12.3 ± 2.3	12.6 ± 1.8	<0.001 *
Protein/day, g/100 kcal ± SD	3.9 ± 0.9	4.2 ±0.8	<0.001 *
Fiber/day, g/100 kcal ± SD	0.8 ± 0.3	0.9 ± 0.3	<0.001 *
Polyunsaturated/saturated fatty acids ratio ± SD	0.7 ± 0.3	0.8 ± 0.3	0.001 *
Sodium per day, mg	3389.2 ± 1272.3	3282.3 ± 1024.4	<0.001 *
DASH score (0–9)	2.66 ± 1.15	3.33 ± 1.12	<0.001 *
Cardiometabolic parameters			
BMI, kg/m^2^	28.5 ± 5.8	27.8 ± 5.1	<0.001 *
HbA1c, %	5.6 ± 0.8	5.5 ± 0.6	<0.001 *
Glucose, mg/dL	97.9 ± 28.4	96.0 ± 22.4	0.001 *
Systolic BP, mmHg	122.8 ± 15.7	120.5 ± 13.9	<0.001 *
Diastolic BP, mmHg	71.2 ± 10.1	70.2 ± 8.4	<0.001 *
HDL, mg/dL	51.9 ± 13.6	56.7 ± 12.6	<0.001 *
LDL, mg/dL	116.5 ± 29.5	115.5 ± 25.5	0.398
Triglycerides, mg/dL	139.1 ± 100.1	121.1 ± 61.5	<0.001 *

BMI: body mass index; BP: blood pressure; HDL: high-density lipoprotein; LDL: low-density lipoprotein; n.a.: Not applicable; * statistically significant. # See methods regarding physical activity level definition.

**Table 2 nutrients-11-01482-t002:** Odds ratios of disease in subjects exposed to probiotics compared to non-exposed subjects.

	Unadjusted, OR (95% CI)	*p* Value	Model 1, OR (95% CI)	*p* Value	Model 2, OR (95% CI)	*p* Value
**Obesity**	0.78 (0.71–0.86)	<0.001	0.82 (0.75–0.90)	<0.001	0.83 (0.76–0.92)	<0.001 *
**Diabetes**	0.84 (0.73–0.97)	0.020	0.96 (0.82–1.13)	0.650	0.97 (0.81–1.17)	0.783
**Hypertension**	0.88 (0.81–0.96)	0.004	0.76 (0.68–0.84)	<0.001	0.79 (0.71–0.89)	<0.001 *
**Dyslipidemia**	0.93 (0.85–1.02)	0.141	0.95 (0.86–1.05)	0.356	1.01 (0.90–1.13)	0.863

Model 1: Age, sex, race, income and education; model 2: Model 1 + alcohol intake, smoking status, physical activity, kcal per day, carbohydrates/kcal per day, protein/kcal per day, fiber/kcal per day, and polyunsaturated/saturated fatty acids ratio. Model 2 also includes BMI in all analyses except in the obesity analysis and includes sodium intake per day only in the hypertension analysis. * Statistically significant.

**Table 3 nutrients-11-01482-t003:** Variation of cardiometabolic parameters in participants exposed to probiotics compared to non-exposed participants.

	Unadjusted	*p* Value	Model 1	*p* Value	Model 2	*p* Value
BMI, kg/m^2^	−0.74 (−1.01 to −0.46)	<0.001 *	−0.47 (−0.75 to –0.20)	0.001 *	−0.41 (−0.67 to −0.15)	0.002 *
HbA1c ^a^, %	−0.03 (−0.05 to −0.01)	0.003 *	−0.01 (−0.03 to 0.01)	0.382	0.01 (−0.01 to 0.03)	0.590
Glucose ^a^, mg/dL	−0.94 (−1.71 to −0.18)	0.016 *	−0.55 (−1.29 to 0.19)	0.146	−0.17 (−0.90 to −0.55)	0.641
Systolic BP ^b^, mmHg	−2.43 (−3.33 to −1.53)	<0.001 *	−1.99 (−2.83 to −1.16)	<0.001 *	−1.48 (−2.31 to −0.66)	<0.001 *
Diastolic BP ^b^, mmHg	−0.92 (−1.46 to −0.38)	0.001 *	−1.13 (−1.71 to −0.55)	<0.001 *	−0.86 (−1.45 to −0.27)	0.005 *
LDL ^c^, mg/dL	0.03 (−2.53 to 2.59)	0.980	−0.93 (−3.50 to 1.63)	0.472	−0.02 (−2.59 to 2.55)	0.988
HDL ^c^, mg/dL	4.93 (4.10 to 5.75)	<0.001 *	1.89 (1.12 to 2.66)	<0.001 *	1.43 (0.69 to 2.17)	<0.001 *
Triglycerides ^c^, mg/dL	−16.82 (−22.64 to −10.99)	<0.001 *	−11.74 (−18.14 to −5.33)	<0.001 *	−8.52 (−15.18 to −1.86)	0.013 *

Model 1: Age, sex, race, income and education; model 2: Model 1 + alcohol intake, smoking status, physical activity, kcal per day, carbohydrates/kcal per day, protein/kcal per day, fiber/kcal per day, and polyunsaturated/saturated fatty acids ratio. Model 2 also includes BMI in all analyses except in the BMI analysis and includes sodium intake per day only in the BP analyses. ^a^ Excluding participants treated with anti-hypertensive drugs. ^b^ Excluding participants treated with anti-dyslipidemic drugs. ^c^ Excluding participants treated with antidiabetic drugs. * Statistically significant.

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
