# Peer review of "Probiotic Ingestion, Obesity, and Metabolic-Related Disorders: Results from NHANES, 1999–2014"

_nutrients, 2019, doi:10.3390/nu11071482_

Round 1

Reviewer 1 Report

The manuscript describes the relation between the use of probiotics and some health parameters (related to metabolic disturbances), it is well written; nevertheless there is a medular points to consider:

Avoid the use of yogurt as synonym of probiotics: Even when the authors  did not find significant differences in associations according to the source of exposure to probiotics; the yogurt ingest shouldn't be considered as ingest of probiotics, because although most yogurts on the market today have not been heat treated after fermentation others are heat treated killing the probiotics. 

Author Response

Dear Editors,

Please find in attachment the answers and amendments according to the reviewers comments. The changes were made in the manuscript according the suggestions of the reviewers and we answered each individual comment. The changes were highlighted along the manuscript.

REVIEWER 1

We appreciate your suggestions and comments and have revised the manuscript, according to your points.

Point 1: The manuscript describes the relation between the use of probiotics and some health parameters (related to metabolic disturbances), it is well written; nevertheless there is a medular points to consider:

Avoid the use of yogurt as synonym of probiotics: Even when the authors did not find significant differences in associations according to the source of exposure to probiotics; the yogurt ingest shouldn't be considered as ingest of probiotics, because although most yogurts on the market today have not been heat treated after fermentation others are heat treated killing the probiotics. 

Response: We thank the reviewer for highlighting the need to clarify this point. We have clarified probiotics definition in methods section and discussed the possible differences in the composition of probiotics, according to the source of exposure - supplements or yogurts, in discussion section – see lines 537-549 please. We have also changed “probiotic exposure” to “probiotic supplements and yogurt exposure” during some relevant parts of the paper.

Reviewer 2 Report

The article shows results obtained in  cross-sectional study using data comigng from NHANES (USA) from 1999 to 2104. The results point to a relationship between probiotic consumption and metabolic-relateed disorders status in USA. However, there are specific comments:

Major comments

This article provides moderate information to the scientific community, as there are similar articles made, and this also shows a statistical study in which some important parameters are not taken into account (see below)

Sports practice (at least the level of practice as low, moderate and high) has not been considered within the study (it does not appear). This parameter should be included in the study within models 2 and 3, as it can dramatically affect the different comorbidities analysed.  If it is not possible to add it, it should be explicitly indicated that this parameter was not available, and taken into consideration within the discussion and analysis of strengths and weaknesses of the study.

I think it is necessary to at least refer and comment Model 3 analysis. It is true that no more info is added with this study, but if it is included in the paper,  you must consider the results obtained (minimum M&M and results). 

In M&M section, The wording of section 2.5. needs to be modified to improve its understanding. The second paragraph is very long and mixes many concepts. You would need to add some kind of graph or explanatory table about the different models and levels considered in the supplementary material, as well as improve its wording.

In the case of Table 1, it seems that there is a bias with ethnies included in the study (main non-hispanic white). It needs to be discussed during the paper and, if there are, previous work analysing gut microbiome differences in different etnies

Line 151. Dyslipidemia % does not correlate with the results shown in Figure 1. Please specify which is the correct one.

Line 192. Please say something more in the case of diabetes ODDs

Discussion: obesity discussion needs to include that the exercise has not been taken into account.

Discussion: in diabetes, please mention and discuss Odds differences in probiotic and/or yogurt (even authors considered not important, at least it is necessary to discuss it with 1-2 sentences)

Minor comments

Line 54. In probiotic definition, please change "living bacteria" to living microorganisms" as it states in FAO/WHO and Hill et al.

Line 113 to 115. Please modify writing to not repeat "to asses".

Line 129- Please delete one "including" word.

Line 142. The % of females and age are included in Table 1, and also explained below. Please delete the sentence "Among those exposed...." as it is redundant.

Table 1. PLease correct the text of the table to have the rows in a correct and easily readable way

Table 1. What happened with other ethnies? not significant? 

Table 1. It appears in footnotes *referring to statistically significant, but * does not appear in the Table.

In section 3.2. the authors provide the same results as in the figure without filtering or highlighting any. It would be desirable to highlight the differences found and not just duplicate data. 

I think there are also food products included (juice, milk...)please specify in M&M that some food products are considered in food supplement list

Please correct all the Tables, Some data are lost at the right margin

Line 206-207. PLease add "... metabolic distrubances by usind unadjusted model"

Line 248: Please change "was responsible" to contributed, or "was associated to..."

Line 290 and 293: Please change manipulation to "modulation"

Author Response

Dear Editors,

Please find in attachment the answers and amendments according to the reviewers comments. The changes were made in the manuscript according the suggestions of the reviewers and we answered each individual comment. The changes were highlighted along the manuscript.

REVIEWER 2

We appreciate your suggestions and comments and have revised the manuscript, according to your points.

Major comments

This article provides moderate information to the scientific community, as there are similar articles made, and this also shows a statistical study in which some important parameters are not taken into account (see below)

Point 1: Sports practice (at least the level of practice as low, moderate and high) has not been considered within the study (it does not appear). This parameter should be included in the study within models 2 and 3, as it can dramatically affect the different comorbidities analysed.  If it is not possible to add it, it should be explicitly indicated that this parameter was not available, and taken into consideration within the discussion and analysis of strengths and weaknesses of the study.

Response: We agree with the reviewer that physical activity may be an important confounder in our analysis. We have updated our analysis to also include physical activity in model 2 and 3. As physical activity was measured differently along NHANES cycles, we chose to use variables that allowed categorization of physical activity level into three categories (low, intermediate, and high), to combine them into a single variable. This new analysis did not qualitatively impact the associations of probiotics and yogurts with the metabolic variables evaluated. All tables and figures were updated to include the new analyses. The following sentences were added to the methods regarding the evaluation of physical activity (lines 146-156):

- “Physical activity was measured differently along NHANES cycles. We classified participants using variables that allowed categorization of physical activity level into three categories (low, intermediate, and high). From 1999 to 2006 the physical activity level was assessed with the question "compare activity with others of the same age" (participants answering “less active” were classified into category “low”, “about the same” into category “intermediate”, and “more active” into category “high”). From 2007 to 2014 the weekly metabolic equivalents (MET) minutes of physical activity (accounting for vigorous work-related activity, moderate work-related activity, walking or bicycling for transportation, vigorous leisure-time physical activity, and moderate leisure-time physical activity) was divided into tertiles (participants were classified as “low” if included in the lower MET-minute tertile, as “intermediate” if in the middle MET-minute tertile, and as “high” if in the higher MET-minute tertile).”

Point 2: I think it is necessary to at least refer and comment Model 3 analysis. It is true that no more info is added with this study, but if it is included in the paper, you must consider the results obtained (minimum M&M and results). 

Response: We agree with the reviewer and have updated methods and results section. The following sentences were added to the methods “As a supplementary analysis we performed an additional model (model 3) that classified the diet pattern using the DASH score. In model 3 we included all model 1 covariates plus alcohol intake, smoking status, physical activity, and DASH dietary pattern score. Model 3 also included BMI in all analysis except in the obesity analysis; and sodium intake per day only in the hypertension and blood pressure analyses”. Regarding results it was added: “Table S3 and S4 summarize the odds ratio of disease and the variation of cardiometabolic parameters, respectively, according to probiotic supplements or yogurts exposure, after accounting for DASH dietary pattern score. The associations of probiotic supplements or yogurt ingestion with cardiometabolic parameters were not different, after adjusting for DASH diet adherence.”

Point 3: In M&M section, The wording of section 2.5. needs to be modified to improve its understanding. The second paragraph is very long and mixes many concepts. You would need to add some kind of graph or explanatory table about the different models and levels considered in the supplementary material, as well as improve its wording.

Response: We thank the reviewer for highlighting the need to clarify this section. We rephrased the section 2.5 and the legend of all tables and figures that included the description of models. Furthermore, as suggested, we added an explanatory table about the different models in the supplementary material.

Point 4: In the case of Table 1, it seems that there is a bias with ethnies included in the study (main non-hispanic white). It needs to be discussed during the paper and, if there are, previous work analysing gut microbiome differences in different etnies

Response: We added the % of other ethnic groups, according to probiotics exposure, in table 1. We also have discussed the impact of gut microbiome differences by geographic ancestry in the extrapolation of our results to other ethnic groups. Please see lines XXXX

Point 5: Line 151. Dyslipidemia % does not correlate with the results shown in Figure 1. Please specify which is the correct one.

Response: The percentage of people exposed to probiotics with dyslipidemia was not correct in the sentence. The correct percentage is 52.3%.

Point 6: Line 192. Please say something more in the case of diabetes ODDs

Response: We have added a new sentence highlighting the odds of diabetes in according to yogurts consumption and probiotic supplements (lines 358-361):

- “Although the association in both analyses was not significant, the odds ratio for diabetes was 0.92 (95%CI 0.77-1.10) among participants exposed to yogurts, and 1.28 (95%CI 0.71-2.23) among those exposed to probiotic supplements.”

Point 7:  Discussion: obesity discussion needs to include that the exercise has not been taken into account.

Response: We have now included this in the discussion. Please see lines 432-435 and 519-521

Point 8: Discussion: in diabetes, please mention and discuss Odds differences in probiotic and/or yogurt (even authors considered not important, at least it is necessary to discuss it with 1-2 sentences)

Response: We have now included this the discussion. The following sentences were added (lines 462-467):

- “In the analysis according to the origin of probiotics, we found a non-significant trend to higher odds of diabetes in participants exposed to probiotics supplements, which was not seen in participants exposed to yogurt consumption. This may be explained by a reverse causality relationship. Participants with diabetes may be more prone to consume probiotics supplements due to their known potential to improve glucose control in diabetes.

Minor comments

Point 1: Line 54. In probiotic definition, please change "living bacteria" to living microorganisms" as it states in FAO/WHO and Hill et al.

Response: We have corrected. See line 80, please

Point 2: Line 113 to 115. Please modify writing to not repeat "to asses". FEITO

Response: We have corrected. See line 165, please

Point 3: Line 129- Please delete one "including" word.

Response: We have corrected. See line 181, please

Point 4: Line 142. The % of females and age are included in Table 1, and also explained below. Please delete the sentence "Among those exposed...." as it is redundant.

Response: We have corrected. See line 194, please

Point 5: Table 1. PLease correct the text of the table to have the rows in a correct and easily readable way

Response: We have corrected. See table 1, please

Point 6: Table 1. What happened with other ethnies? not significant? 

Response: We added the % of other ethnic groups, according to probiotics exposure, in table 1.

Point 7: Table 1. It appears in footnotes *referring to statistically significant, but * does not appear in the Table.

Response: We have corrected. See table 1, please

Point 8: In section 3.2. the authors provide the same results as in the figure without filtering or highlighting any. It would be desirable to highlight the differences found and not just duplicate data. 

Response: We have changed the text to highlight the main differences, as suggested. See section 3.2, please.

Point 9: I think there are also food products included (juice, milk...)please specify in M&M that some food products are considered in food supplement list

Response: The reviewer is correct. Non yogurt foods containing probiotics were classified as probiotic supplements. We have now included this information in M&M.

Point 10: Please correct all the Tables, Some data are lost at the right margin

Response: We have corrected all the tables.

Point 11: Line 206-207. PLease add "... metabolic distrubances by usind unadjusted model"

Response: We have corrected. See lines 250-251, please

Point 12: Line 248: Please change "was responsible" to contributed, or "was associated to..."

Response: We have changed. See line 492, please

Point 13: Line 290 and 293: Please change manipulation to "modulation"

Response: We have changed. See line 554 and 557, please

Reviewer 3 Report

Novelty of investigation need to mention in abstract and introduction. 

The composition of intestinal microorganism in obesity, and metabolic-related disorders is need to mention in introduction.

Section 2.2 need to improve. However, authors mentioned about number of initial members, placebo group and experimental group, criteria for inclusion and exclusion need to mention in clear way. How many members finally finished study protocol and appear to interview need to describe. It is better to describe the method by flow chart.

What was dose of probiotics and youghurt ? What was the yoghurt composition ? - need to mention 

The preventive mechanisms of Obesity, Diabetes, Hypertension and Dyslipidemia by probiotics need to mention in discussion section.

Author Response

Dear Editors,

Please find in attachment the answers and amendments according to the reviewers comments. The changes were made in the manuscript according the suggestions of the reviewers and we answered each individual comment. The changes were highlighted along the manuscript.

REVIEWER 3

We appreciate your suggestions and comments and have revised the manuscript, according to your points.

Point 1: Novelty of investigation need to mention in abstract and introduction. 

Response: We thank the reviewer for highlighting this point. We have changed the abstract and introduction, in order to emphasize the novelty of investigation. Please see changes in abstract and introduction.

Point 2: The composition of intestinal microorganism in obesity, and metabolic-related disorders is need to mention in introduction.

Response: We have now included this in the introduction. Please see changes in introduction section.

Point 3: Section 2.2 need to improve. However, authors mentioned about number of initial members, placebo group and experimental group, criteria for inclusion and exclusion need to mention in clear way. How many members finally finished study protocol and appear to interview need to describe. It is better to describe the method by flow chart.

Response: We thank the reviewer for highlighting the need to improve the section 2.2. We clarified the inclusion and exclusion criteria (see lines 110-112, please) and we now present in supplementary material the flowchart of the study population.

Point 4: What was dose of probiotics and youghurt ? What was the yoghurt composition ? - need to mention 

Response: We have clarified the definition of probiotics exposure (methods section) and have discussed this point in the discussion section. Please see lines 130-132 and 537-549.

Point 5: The preventive mechanisms of Obesity, Diabetes, Hypertension and Dyslipidemia by probiotics need to mention in discussion section.

Response: We thank the reviewer for highlighting this point. We have added this point in the discussion section. Please see lines 409-427.

Round 2

Reviewer 1 Report

The authors let clear the difference between probiotics and source or probiotics in the abstract 

"ingestion of probiotic supplements or yoghurts " (abstract) 

but the sentence in the page 3, line 104 is not clear enough, 

"Non yogurt food containing probiotics were classified as probiotic supplements"  (Why would be classified as probiotic supplements? It would be better use yogurt as (dietary) probiotics source...) 

Are the authors sure that all the yogurts contained probiotics? Let clear in the manuscript that yogurts are one of the sources of probiotics... 

Author Response

Dear Editors,

Please find in attachment the answers and amendments according to the reviewers comments. The changes were made in the manuscript according the suggestions of the reviewers and we answered each individual comment. The changes were highlighted along the manuscript. 

REVIEWER 1

We appreciate your comments and have revised the manuscript, according to your points.

Point 1: The authors let clear the difference between probiotics and source or probiotics in the abstract ("ingestion of probiotic supplements or yoghurts "), but the sentence in the page 3, line 104 is not clear enough ("Non yogurt food containing probiotics were classified as probiotic supplements". Why would be classified as probiotic supplements? It would be better use yogurt as (dietary) probiotics source...

Are the authors sure that all the yogurts contained probiotics? Let clear in the manuscript that yogurts are one of the sources of probiotics...

Response: We thank again the reviewer for highlighting the need to further clarify this point. As suggested, we clarified that yogurts are one of the sources of probiotics,  in line 106 “…yogurt (as a dietary source of probiotics)…”.

Previously, we have clarified probiotics definition in methods section and discussed the possible differences in the composition of probiotics, according to the source of exposure - supplements or yogurts, in ‘Discussion’ section. We have also submitted a comprehensive supplementary table of all the probiotic containing food categorized in NHANES. After some reviewers’ suggestions we decided to clarify and clearly separate the: 1) yogurts (as source of probiotics), and 2) probiotic supplements (including all the products with the addition of probiotic). In this last category, any food that was actively added with probiotics was considered. Many ingestion products can serve as probiotic-added products, like pills, juice or any kind of food.

Reviewer 2 Report

The article now is ok for being published

Author Response

We appreciate all your comments and suggestions and have revised the manuscript, according to your points.

Reviewer 3 Report

I read the full manuscript in pdf version and checked the response, written by authors. I am agree with the response, provided by authors.

I recommend this article for acceptance.

Author Response

We appreciate all your comments and suggestions and have revised the manuscript, according to your points